# Localized Floods, Poverty and Food Security: Empirical Evidence from Rural Pakistan

**Akhter Ali [1],\* and Dil Bahadur Rahut [2]** 

[1]   International Maize and Wheat Improvement Center (CIMMYT), CSI Complex, NARC, Park Road, Islamabad 44000, Pakistan
[2]   International Maize and Wheat Improvement Center (CIMMYT), El Batan, Texcoco C.P. 56237, Mexico; d.rahut@cgiar.org
\*   Correspondence: akhter.ali@cgiar.org

**Abstract:** National level floods affect large sections of the population, and in turn, receive attention from the government and international agencies. Localized natural disasters, including localized floods, do not get the attention of the government and policymakers because their impact is felt within limited geographical areas, despite the fact that these disasters severely affect the livelihood of rural communities. This study examines the impact of localized floods on the livelihood of farmers in Pakistan using a cross-sectional data set collected from 812 households. The empirical results show that localized floods severely affect rural livelihoods, and affected households have lowered cereal crop yields, less income, and reduced food security levels. Farmers adopt a number of strategies, including crop and livestock insurance, bund-making, land-leveling, and tree planting, to combat the impact of localized floods. Among all these mitigating strategies, the tree plantation is ranked as the best mitigating strategy followed by crop and livestock insurance, land leveling, and bund making, respectively. Education, wealth, access to non-governmental organizations (NGOs), extension services, and infrastructure, influence the adoption of measures to mitigate the effect of flood risks. National policy on localized flood risks needs to strengthen local institutions to provide support to families and extension services to train farmers to mitigate the impact of localized floods.

**Keywords:** localized floods; rural livelihood; flood risk management; Pakistan

## 1. Introduction

Floods are natural hazards which have severe adverse consequences on property, infrastructure, livelihood, livestock, human lives, human health, the environment, cultural heritage, and economic activity. Across the globe, annual flooding leads to the loss of about 20,000 lives and negatively impacts 20 million people [1], which is predicted to rise during the next decade [1]. It has adversely affected over 2.8 billion people since 1990 across the world [2]. Losses of life and property due to flood events is largely due to inadequate structural and non-structural measures, and incompetent and inefficient policy responses [3,4].

In Pakistan, the floods of 2010 and 2013 caused a lot of damage [5–7]. Floods generally cause damage to homes and businesses situated in the natural floodplains of rivers [8]. Floods also result in the displacement of people, infrastructure damage (such as destruction of roads), loss of crops, cattle, and livestock and these losses delay ongoing development and political processes [9,10] which in turn have serious implications for food security, particularly in poor and developing countries [9].

Quickly draining the flood water or storing it temporarily and constructing dams and walls to protect life and property was the conventional approach to flood-risk management. However, under

an integrated flood-management approach, land, and water resources in the river basin are managed judiciously to make the most efficient use of floodplains and reduce the loss of life and property [11].

Previous research results show that increasing tree cover has a small but statistically significant effect on reducing channel discharge [12]. Planting trees reduces exposure to the flood, thereby reducing damage from the flood. As the risk is a function of hazard, vulnerability, and exposure, the damage from the flood is generally proportional to the reoccurrence period [13].

Experience with the risk and strategies to manage the risk affects the perception towards risk, which generally influences the resilience and the risk-mitigating strategies adopted by the household and community. The ability of a community to mitigate, manage with, and recuperate from adverse effects of floods depends on the perception and impression of the community about events [14].

Pakistan is a flood-prone country with a history of widespread and repeated flooding that causes loss of lives, substantial damage to property, infrastructure, loss of agricultural crops, and land [15]. It has two dominant types of floods: riverine and flash.

Flooding is a frequently occurring and catastrophic natural threat in Pakistan, but the nature of the flood varies geographically [16]. Table 1 shows that a 2010 flood disaster was massive, killing more than 1700 people, affecting over 20% of the land area and more than 20 million people, and causing billions of dollars of loss through damages to infrastructure, housing, agriculture and livestock, and other family assets [17]. It destroyed more than two million homes, two million hectares of standing crops, and 1.2 million head of livestock. The overall loss of sugar cane, paddy, and cotton was estimated at 13.3 million MT [18].

**Table 1.** Incidence of floods in Pakistan during the last two decades.

| Flood Year | People Affected | Lives Lost | Reference/Source |
|---|---|---|---|
| 2003 | 4376 | 484 | Pakistan Weather Portal |
| 2007 | 2 million | 918 | Pakistan Weather Portal |
| 2010 | 20 million | <1781 | Rehman et al., (2015) and Pakistan Weather Portal |
| 2011 | 8.9 million | 434 | Pakistan Weather Portal |
| 2015 | 2.5 million | 367 | Pakistan Weather Portal |

Source: Website Report (https://pakistanweatherportal.com/).

The recent flood of September 2015 hit all of Pakistan, resulting in the deaths of 367 individuals. Over 2.5 million people were affected by the floods and rains, and 129,880 houses were partially damaged or entirely destroyed [19]. The estimated recovery effort was about US$439.7 million. The agricultural sector was the most affected: around one million acres of standing crops were destroyed, and 250,000 farmers were affected [19]. As agriculture contributes about 21% to Pakistan's GDP, 43% to employment, and 60% of the export earnings [19], the adverse impact of flooding on the agricultural sector is likely to severely affect livelihoods, poverty and the economy of Pakistan.

Against the backdrop of the frequency of floods and its devastating impact on the economy and lives of the people in developing countries, it is important to understand the impact of floods, particularly localized floods that do not reach to the attention of the national government and international agencies, as well as the mechanisms that households adopt to manage localized flood risk. The contribution of this paper is twofold: firstly to the best of our knowledge this is the first paper that assesses the household-level impact of localized floods; secondly using primary household-level data, it explores the strategies adopted by rural farm households to mitigate the effect of localized floods and the factors influencing this adoption. In this paper, the impact of localized floods is estimated on crop yield, household income, and food security levels. A comprehensive cross-sectional data set was collected from 812 households. The data was collected from all the four provinces of Pakistan: Punjab, Sindh, Khyber Pakhtunkhwa (KP), and Balochistan. The empirical analysis was carried out by employing the propensity score matching approach.

The rest of the paper is organized as follows: in Section 2, brief review of literature is provided, in Section 3 materials and methods are outlined, in Section 4 results and discussion are presented, and in Section 5 the conclusion along with some policy recommendations is presented.

## 2. Review of Literature

The frequency of occurrence of floods in the region in general and Pakistan, in particular, has considerably increased in the past several years because of rapid climate change [20,21]. Because of this, Pakistan has faced a series of flood events since 2010, indicating that floods are now a regular occurrence in the country. Today, flooding affects people across the globe and particularly in Bangladesh, India, and Pakistan, more frequently than before [20,21]. Since 1960 Pakistan has suffered a cumulative financial loss of more than US$ 38.17 billion, approximately 12,330 people have lost their lives, some 197,275 villages have been damaged or destroyed, and an area more than 616,598km$^2$ has been affected due to 24 major flood events [22].

There are two main categories of floods: natural floods, which are usually due to heavy rainfall and snowmelt and floods due to human activities, such as the failure of dikes and other protective structures. Floods may result from the overflowing of a vast body of water over land and extreme hydrological events or an unusual presence of water on land to a depth which affects normal activities [23]. They also occur as a result of climatological and hydrological extremes as well as human activities on drainage basins [24]. Due to changing climatic conditions, natural disasters, particularly flooding, are occurring quite frequently [25], which can cause serious damage in the form of infrastructure destruction, human and livestock casualties as well as damage to crops.

Floods also lead to health problems. For example, a study found that floods cause extensive disease and mortality across the world, and the impact of floods on the community is dependent on the topography of the area and its human demographics [26]. Lack of understanding and preparedness exacerbate the adverse impact of floods, particularly among poor and illiterate families in developing countries.

As the risk is a function of hazard, vulnerability, and exposure, the damage of the flood is generally proportional to the reoccurrence period [13]. Flood risk-mitigation measures should, therefore, firstly focus on reducing exposure to floods by building dykes or dams, building terraces, or planting trees. Secondly, it should focus on reducing vulnerability and building resilience through insurance, early warning systems, government support mechanisms, and community preparedness.

As floods have severe consequences in the lives and livelihoods of people living in the fragile areas of developing countries [10], the adoption of integrated and economical approaches to mitigate the effects of floods is crucial. It is of paramount importance for individuals and the government to prepare for these unforeseen circumstances.

Studies have found that individuals' perception of flood risk is one of the critical factors influencing the adoption of non-structural strategies to mitigate the impact of flooding [27]. The perception and impression of the flood also help the community recuperate from the adverse effects of floods [14]. If an individual perceives flooding as a risky event, it is highly likely that an individual will adopt flood risk mitigation measures [28,29]. However, various research found a statistically weak relationship between risk perception and mitigation behaviours [27]; this may be because, after implementation of the flood risk-mitigation measures, risk perception decreases [30,31].

Planting trees helps to protect the farm and property of the household by decreasing the exposure to and reducing the intensity of the flood. From 7 eligible studies of 156 papers reviewed, results show that increasing tree cover has a small statistically significant effect on reducing channel discharge [12]. Therefore, reducing deforestation and propagating aforestation seems to be a cheap and soft non-structural adaptation mechanism to combat flooding.

Insurance is also an emerging option for adaptation against flood risks. Well designed insurance could play a significant role in mitigating the adverse effect of the flood and protect livelihoods and communities [32]. Although insurance is a useful instrument to cope with risk, individuals

are sometimes reluctant to purchase any if the cost of insurance is higher than the flood risk [33]. Besides providing financial protection against the risk of flooding, insurance can also play a role by encouraging prevention, preparedness, and response measures [34]. At times flood risk insurance may act as moral hazard and increase exposure to risk [16,35]. For example, as an individual can get new items for the damaged items, they have less incentive to protect items or reduce exposure to the risk [36]. In order for flood risk insurance to be incorporated into flood risk-management practices, it is essential that the dweller of the floodplain find this option acceptable [16]; awareness of risks, appropriate premiums, affordability, and backing by the government may enhance the acceptability of insurance as a risk-mitigation strategy.

Following some renewed attention on non-structural flood risk mitigation measures implemented at the household level, there has been an increased interest in the socio-economic and perceptual factors that influence precautionary behavior [37–40]. Pakistan's approach to flood management planning is found to be largely inadequate, and this inadequacy is mainly attributed to missing links in policy formulation and planning processes, along with a lack of institutional coordination [41]. The existing flood management framework in Pakistan underestimates the importance of community participation in flood risk-mitigation planning [21]. Pakistan, despite being the second largest recipient of adaptation funding after Bangladesh, is one of the least adaptive countries in the world.

There is a lack of effective coordination among institutions involved in flood management, caused in part by limitations of technical capacities such as dissemination of early warnings, disaster preparedness measures, emergency response, and structural measures for flood mitigation. In addition, local communities do not have enough disaster preparedness information, and there is a lack of awareness-raising, sensitization, and education of the population regularly affected by floods.

The damage resulting from national-level floods has been described and discussed by a number of researchers, but the impact of localized floods have not been well documented, and they go unnoticed in the research. Although localized floods occur quite frequently in rural areas and adversely impact the lives and livelihood of rural farm households, they do not catch national-level attention, as the geographical spread is smaller, and the level of intensity is lower. As a result, farmers do not receive any post and ex-ante support to cope with localized floods. Although a number of studies have focused on the impact of climate change and environmental hazards, there are hardly any studies which have focused on localized floods or limited-scale natural disasters, which have a significant impact on rural communities. Although we cannot stop the flood, we could put measures in place to mitigate the impact of floods and provide support to those families affected. Against this backdrop of losses and the need for assessing the impact, this paper attempts to assess the impact of the localized flood and the adaptation strategies adopted by farm households to cope with flood shocks.

## 3. Materials and Methods

### 3.1. Conceptual Framework

Although localized floods are small in terms of geographical area affected and limited in intensity of effects, they disrupt the livelihoods of rural farm households, which can be devastating at times. Localized floods adversely affect several farm households in a community, thereby reducing the yield of rice, maize, and wheat and, therefore, income, and increases the poverty level.

As localized floods affect farm households negatively, they adopt several mechanisms to mitigate the negative impact of flood risk. Farm households in Pakistan generally adopt four major strategies to mitigate the impact of the flood risk: crop and livestock insurance, bund-making, land-leveling, and tree plantation. Several socioeconomic and demographic factors influence the farm household's decision to adopt measures that mitigate the adverse effects of flooding.

We consider two categories of farmers: those who adopt different localized flood-management practices and those who do not adopt any flood-management strategies. Households adopt measures

to mitigate flood risk only if the utility derived from the adoption of the flood risk strategies is higher than from not adopting the mitigation measure.

$$U(\psi_i, \Phi_1, \Omega_2, \Gamma_3, \eta_4) > U(\psi_1, \Phi_1, \Omega_2, \Gamma_3) \tag{1}$$

In Equation (1), $\Phi_1$ is the crop and livestock insurance, $\Omega_2$ is the bund-making strategy, $\Gamma_3$ is the laser land-leveling technology $\eta_4$ is the tree plantation done to manage localized floods. Equation (1) indicates that households having adopted four flood risk-management strategies have higher utility levels compared to households that have adopted only three or fewer flood risk-management strategies.

These higher utility levels can translate into fewer yield losses, higher household income levels, and greater food security levels.

### 3.2. Data and Sampling

Data was collected through detailed field surveys from 812 households across Pakistan in 2017. The distribution of the sample by province is provided in Table 2. In the first stage, we selected all five provinces of Pakistan, and in the second stage, we selected different Tehsil, and finally, we selected farm households. Well-trained enumerators conducted the survey. Prior to the implementation of the survey, we conducted a pilot survey. Based on the feedback received from the pilot survey, we revised the instrument. During the survey, information on a number of households and farm-level characteristics was collected. Specifically, we collected information on the occurrence of floods during the last five years. The description of the sample is presented in Figure 1.

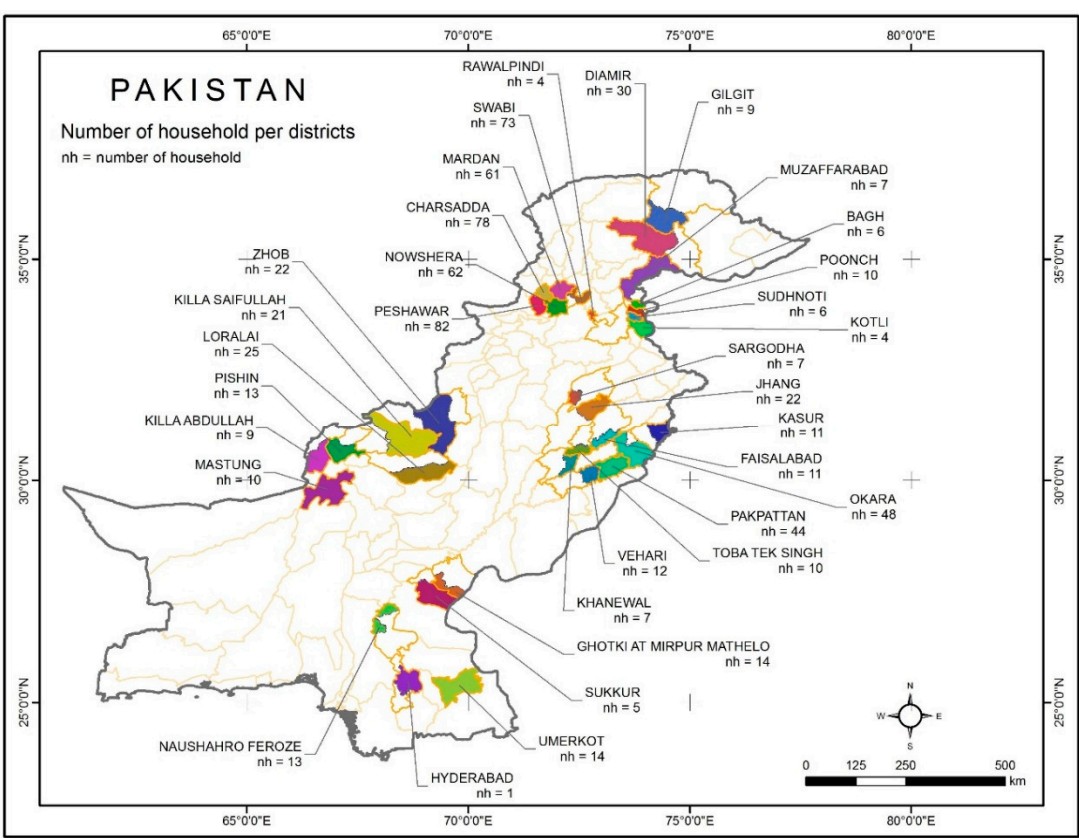

**Figure 1.** Description of the sample.

**Table 2.** Sample of Tehsil/village.

| Province | Tehsil/Village |
|---|---|
| Sindh (51) | Khan Garh, Hyderabad, Matiari, Kandiyaro, MoroRohri, Saleh Pat, Tando Allah yar, Kunri, Pithoro, Summaro, Umar Kot |
| Azad Jammu and Kashmir (AJK) (53) | Bagh, Hattian Bala, Kotli, Muzaffarabad, Aath Maqam, Hajeera, Rawalakot, Palandri, Sadhnoti, Tarar khal |
| Gilgit Baltistan (45) | Gilgit, Ali Abad, Sikandarabad, Chillas, Darail, Tangir |
| Punjab (207) | Kasur, Chinot, Bowana, Behra, Taxila, Shalimar, Okara, Depalpur, Tandalainwala, Sahiwal, Pakpatten, Arifwala, Burawala, Mian chanu, Kamalia |
| Balochistan (100) | Killa Abdullah, Gulistan, Killa Saifullah, Muslimbagh, Duki, Mekhtar, Bori, Kad Kocha, Dashat, Mastung, Pishin, Huramzai, Karezat, Zhob |
| Khyber Pakhtunkhwa (KP) (356) | Tangi, Charsada, Mardan, Takhtbai, Katlang, Nowshara, Pabi, Peshawar, Topi, Swabi |

Note: In parenthesis, the sample size from each respective region is given.

### 3.3. Multivariate Probit

A multivariate probit model was used to estimate the factors influencing the choice of flood management strategies adopted by farm households in Pakistan.

### 3.4. Propensity Score-Matching (PSM) Approach

For the estimation of the impact, we used propensity score matching (PSM). The significance of the PSM arises from the fact that it accounts for sample selection bias when experimental data are not available [42]. In the absence of experimental data, PSM creates the condition of the randomized experiment [43].

The advantage of the PSM, as opposed to the parametric approach, is that PSM does not entail assumptions about the functional form in stipulating the relationship between outcomes and forecasters of the outcome. The downside of the approach is the heavy assumption of unconfoundness. It may be possible to notice systematic differences between outcomes of adopters and non-adopters even after conditioning because the selection is based on unmeasured features [44]. Nonetheless, the supposition is no more restrictive than those of the instrumental variable approach engaged in cross-sectional data analysis [45]. In this paper, two different matching algorithms, i.e., nearest neighbour matching and kernel-based matching, are employed to check the robustness of the estimates [46,47]. After matching, a number of balancing tests are employed to check the matching quality like median absolute bias before and after matching, the value of R-square, and the joint significance of covariates before and after matching.

## 4. Results and Discussion

### 4.1. Descriptive Analysis

4.1.1. Occurrence of Flood Events during the Last Five Years

Farmers were asked if they experienced localized floods during the last five years, and 82.7% of farmers had not, while 17.3% had (see Table 3). Unlike big national floods, localized floods are experienced only in certain communities and in specific locations and are unlikely to get the attention of the government, leaving people to cope on their own.

**Table 3.** Floods during the last five years.

| Number of Floods during the Last Five Years | Frequency | Percentage |
|:---:|:---:|:---:|
| 1 | 108 | 13.3 |
| 2 | 21 | 2.6 |
| 3 | 5 | 0.6 |
| 4 | 3 | 0.4 |
| 5 | 3 | 0.4 |
| None | 672 | 82.7 |

### 4.1.2. Description of the Variables Used in the Study

The descriptive statistics and the description of variables used in the empirical model is summarized in Table 4. The mean age of the farmers was about 42 years, and 92% were married, while the mean education level was about seven years of schooling, and the mean number of years of experience was 22. This shows that, like many other developing countries, farmers are mostly married and middle-aged individuals. The average years of schooling of seven also indicate that the farmers were fairly educated and capable of learning new tools and techniques.

**Table 4.** Data and description of variables.

| Variable | Description | Mean | Std. Dev |
|:---|:---:|:---:|:---:|
| *Demographic* | | | |
| Age of farmer | Age of the farmer in number of years | 42.65 | 13.09 |
| Education | Education of the farmer in number of years | 6.7 | 5.62 |
| *Land assets* | | | |
| Tenancy status | 1 if the farmer is owner, 0 otherwise | 0.27 | 0.70 |
| Land owned | Number of acres of land owned by the farmer | 51.99 | 19.82 |
| Area type | 1 if the area is irrigated, 0 otherwise | 0.68 | 0.69 |
| Land slope | 1 if slope is same (even land), 0 otherwise | 0.31 | 0.25 |
| Laser land-leveling | 1 if the farmer practiced laser land-leveling, 0 otherwise | 0.45 | 0.46 |
| *Farm assets* | | | |
| Tractor | 1 if the household owns a tractor, 0 otherwise | 0.32 | 0.46 |
| Tube well | 1 if the household owns a tube well, 0 otherwise | 0.35 | 0.47 |
| Mold bold (MB )plough | 1 if the household owns a mould bold plough, 0 otherwise | 0.21 | 0.40 |
| Seed drill | 1 if the household owns a seed drill, 0 otherwise | 0.12 | 0.32 |
| Ridger | 1 if the household owns a ridger, 0 otherwise | 0.18 | 0.38 |
| *Durable assets* | | | |
| Car | 1 if the household owns a car, 0 otherwise | 0.20 | 0.40 |
| Television | 1 if the household owns a television, 0 otherwise | 0.79 | 0.78 |
| Pakistan Telecommunication Company Limited (PTCL) | 1 if the household has access to ptcl, 0 otherwise | 0.09 | 0.29 |
| Tree Plantation | 1 if the farmer has planted trees, 0 otherwise | 0.13 | 0.46 |
| *Access to facilities* | | | |
| Paved road | 1 if the household has access to a paved road, 0 otherwise | 0.62 | 0.45 |
| Basic health unit | Distance to the nearest basic health unit in kilometers | 0.54 | 0.35 |
| Veterinary center | Distance to the veterinary center in kilometers | 0.38 | 0.14 |
| School | 1 if there is a school in the village, 0 otherwise | 0.91 | 0.11 |
| Implement repair | 1 if the household has access to implement repair shop, 0 otherwise | 0.24 | 0.42 |
| Output market | 1 if the household has access to the output market, 0 otherwise | 0.14 | 0.35 |
| Non-governmental organizations (NGOs) | 1 if the household has access to NGOs, 0 otherwise | 0.10 | 0.30 |

About 27% of the farmers owned land, and the rest were tenant farmers. The average landholding of the farmers was about 52 acres, which indicates that the sampled household were medium holder farmers. Only 31% of the farmers reported that the land had the same slope (i.e., even land), which is better for farm management. The majority of farmers (68%) were living in irrigated areas, which means that a large proportion of the sampled households did not have to depend on the monsoon for cropping. The other 32% were living in rain-fed areas. The sampled farmers were using some improved farming practices, as about 45% had practiced land laser-leveling.

Farm assets are essential for farming and play an important role in farm productivity. Summarizing the information on the farm assets highlights that the level of mechanization is comparable

to other south Asian countries. Of the sampled farmers, 32% of them owned tractors, 35% owned tube wells, 21% owned moldbold (MB) ploughs, only 12% have seed drills, 18% owned ridgers, and 4% owned reapers.

Durable assets owned by the farmer reflect their wealth status and capability to cope with localized floods. Only 20% of the farmers owned a car, while 75% owned a motorcycle, and 41% owned a bicycle. Washing machines were owned by 74% of the farmers, 70% of the farmers owned a refrigerator, 14% had air conditioners, 27% had a room cooler, 93% had an iron, and 79% owned a television. Based on the asset ownership, we can confirm that the majority of the farmers had the capacity to cope with the localized flood.

Access to facilities also plays an essential role in managing the risk of localized floods. In the study areas, the data shows that farmers had reasonably good access to facilities and infrastructure. The descriptive analysis shows that 62% of the farmers had access to a paved road. The average distance to a basic health unit was 0.5 km, while the distance to the veterinary center was 0.38 km. A school was available in 91% of the villages; only 17% of the villages had a bank facility, 24% had access to an implement repair shop, and 14% had access to output markets. About 94% of the farmers had access to electricity, 23% to pesticide dealers, 30% had access to water supplies, and 35% had access to a post office. Only 10% of the farmers had NGO memberships.

*4.2. Empirical Results*

4.2.1. Determinants of the Flood-Management Strategies Adopted by the Farmers

We used the multivariate probit model to estimate the factors influencing the choice of flood management strategies adopted by farmers in Pakistan. The dependent variables are the flood management strategies, namely crop and livestock insurance, bund-making, land-leveling, and tree plantation. As farmers adopted more than one adaptation strategy simultaneously, the multivariate probit is the most suited model. The cross-equation correlations confirm the suitability of the model used (see Table 5). The age of the farmer is negative and significant for crop and livestock insurance and land-leveling, while it is positive and significant for bund-making and tree plantation. Younger farmers tended to adopt crop and livestock insurance and land-leveling, but older farmers tended to depend more on bund-making and tree plantation. This is an indication that elderly farmers adopt more traditional flood-management strategies, while younger farmers tend to adopt newer technologies. However, an integrated flood-management approach should focus on the adoption of a wide range of measures, thereby making farmers more resilient to localized flood risks.

Landowners are more likely to adopt crop and livestock insurance, bund-making and land-leveling and are less likely to adopt tree plantation as a strategy to manage localized floods. Similarly, the coefficient of the number of hectares of land owned is positive for crop and livestock insurance, bund-making, and land-leveling, and significant for all except tree plantation, signifying that tree plantation is not considered adequate by landowners and those with large landholdings. The landowner and the farmer with large landholdings have the capacity to purchase insurance and invest in bund-making and land-leveling technology.

The coefficient of the tractor ownership dummy is positive for all four strategies, but significant only for bund-making. Similarly, the ownership of a tube well is positive and significant for bund-making, land-leveling, and tree plantation. Hence, it can be concluded that farmers with more farm assets rarely use crop and livestock insurance to cope with flood risk. The coefficient of the MB plough is negative and significant for crop and livestock insurance, bund-making, and land-leveling, which means farmers with MB plough are poor farmer and are less likely to adopt any of the flood risk management strategies.

**Table 5.** Flood-management strategies adopted by the farmers (multivariate probit estimates).

| Variable | Crop and Livestock Insurance | Bund-Making | Land-Leveling | Tree Plantation |
|---|---|---|---|---|
| *Demographic* | | | | |
| Age | −0.02 **(2.14) | 0.01 ***(3.08) | −0.03 *(1.82) | 0.02 **(2.04) |
| Education | 0.02 ***(2.57) | 0.03(0.85) | 0.01 ***(3.17) | −0.03 *(1.73) |
| *Land assets* | | | | |
| Tenancy status | 0.01 ***(2.52) | 0.01(1.22) | 0.03 **(2.18) | −0.01 *(1.68) |
| Land owned | 0.02 ***(2.73) | 0.03 **(2.28) | 0.01 ***(3.04) | 0.04(1.12) |
| Area type | 0.01(1.32) | 0.02(1.45) | 0.02 **(2.16) | 0.01 *(1.92) |
| Slope | 0.01 **(2.15) | −0.02 *(−1.74) | 0.05 *(1.92) | −0.02 ***(2.84) |
| *Farm Assets* | | | | |
| Tractor | 0.02(1.35) | 0.01 *(1.84) | 0.02(1.61) | 0.01(0.42) |
| Tube well | 0.02(1.29) | 0.03 **(2.13) | 0.01 ***(2.78) | 0.01 *(1.81) |
| MB plough | −0.01 *(−1.83) | −0.02 **(−2.34) | −0.01 ***(−2.52) | −0.02(1.41) |
| *Household assets* | | | | |
| Car | 0.01 *(1.75) | 0.02 *(1.89) | 0.03 **(2.14) | −0.01 *(1.92) |
| Television | −0.02(2.22) | 0.03 **(2.36) | 0.01 ***(2.46) | 0.03 **(2.18) |
| *Access to facilities* | | | | |
| Paved road | 0.02 **(2.11) | 0.03 ***(2.40) | 0.04 **(2.15) | 0.02 *(1.73) |
| NGOs | 0.01**(2.37) | 0.02(1.22) | 0.03(1.36) | 0.01(1.26) |
| *Province* | | | | |
| Punjab | 0.02 *(1.68) | 0.01 **(2.15) | −0.02 *(1.93) | 0.04 *(1.92) |
| Sindh | 0.03 *(1.76) | −0.0 2*(1.66)) | 0.01 *(2.18) | −0.03(−1.53) |
| KP | 0.02(0.99) | 0.01 **(2.14) | 0.03 ***(3.18) | 0.02 **(1.98) |
| Constant | 0.01 **(2.18) | 0.02 **(2.14) | 0.02 *(1.84) | 0.03(1.93) |
| LR $\chi^2$ | 287.36 | | | |
| Prob > $\chi^2$ | 0.000 | | | |
| Value of $R^2$ | 0.27 | | | |
| Cross equation Correlations | $\rho_{12}$ 0.25 ***(2.43) $\rho_{24}$0.20 *(1.72) | $\rho_{13}$ 0.33 ***(2.69) $\rho_{34}$ 0.17(1.28) | $\rho_{14}$ 0.21 ***(2.73) | $\rho_{23}$ 0.18 ***(2.48) |

Note: Results are significant at ***, **, * 1, 5, and 10 percent levels, respectively.

We use several durable assets to identify the role of wealth/assets in the adoption of flood-management strategies. The coefficient of car ownership is positive and significant for crop and livestock insurance, bund-making, and land-leveling, and negative and significant for tree plantation. As only very rich farmers own cars, we found that car-owning farmers tended to adopt techniques requiring large investments, such as crop and livestock insurance, land-leveling, and bund-making. The ownership of a television was positive and significant for bund-making, land-leveling, and tree plantation. Hence, it can be concluded that wealth/assets drive the adoption of flood-management strategies as a higher level of wealth means a greater capacity by the farmer to finance flood management strategies.

Access to facilities and infrastructure is an essential factor determining the adoption of agricultural technology, which is confirmed by the results of our paper. The coefficient of the dummy access to a paved road is positive and significant for all four strategies. Access to facilities and infrastructure provides easy access to technology and knowledge; hence, proximity to facilities enhances the adoption of measures to mitigate the impact of floods. The coefficient of the access to NGOs is positive for all four strategies, but significant only for crop and livestock insurance; this may be because NGOs advocate for crop and livestock insurance. According to farmers among all these mitigating strategies, the tree plantation is ranked as the best mitigating strategy followed by crop and livestock insurance, land leveling, and bund-making, respectively. The adoption of these flood mitigating measures is beneficial to all categories of farmers especially to the small farmers as they are more vulnerable have limited capacity to cope with these challenges.

To control for spatial heterogeneity, we use the provincial dummy; the results show that farmers in Punjab province are more likely to adopt crop and livestock insurance, bund-making, and tree planting and are less likely to adopt land-leveling as a strategy to manage localized floods. Farmers in

KP adopt bund-making, land-leveling, and tree planting, and the farmers in Sindh are more likely to adopt crop and livestock insurance and land-leveling and are less likely to adopt bund-making.

4.2.2. Impact of Localized Floods: Propensity Score Matching

The impact of localized floods was estimated by employing the PSM approach, and the results are reported in Table 6. For PSM, two different matching algorithms are employed viz nearest neighbor matching (NNM) and Mahalanobis metric matching (MMM) to check the robustness of the results across various groups. The impact of localized floods was estimated on the yields of cereal crops like wheat, maize, and rice, and farmer income, poverty, and food security levels. The average treatment effect for the treated (ATT) is the difference in the outcome of similar farmers experiencing localized floods and those not experiencing localized floods.

**Table 6.** Impact of floods on crop yields, household income, food security, and poverty levels.

| Matching Algorithm | Caliper | Outcome | ATT | t-Values | Critical Level of Hidden Bias | Number of Treated | Number of Control |
|---|---|---|---|---|---|---|---|
| NNM | 0.03 | Wheat yield | −2.64 * | −1.75 | 1.25–1.30 | 120 | 145 |
| | 0.05 | Maize yield | −3.11 ** | −2.14 | 1.45–1.50 | 120 | 145 |
| | 0.01 | Rice yield | −2.15 ** | −2.06 | 1.15–1.20 | 120 | 145 |
| | 0.02 | Income | −3045 *** | −3.11 | 1.30–1.35 | 120 | 145 |
| | 0.009 | Poverty | 0.04 | 1.43 | 1.05–1.10 | 120 | 145 |
| | 0.06 | Food security | −0.04 ** | −2.15 | 1.25–1.30 | 120 | 145 |
| MMM | 0.01 | Wheat yield | −2.18 ** | −2.23 | 1.35–1.40 | 143 | 162 |
| | 0.02 | Maize yield | −2.53 ** | −1.98 | 1.20–1.25 | 143 | 162 |
| | 0.03 | Rice yield | −2.07 *** | −3.14 | 1.65–1.70 | 143 | 162 |
| | 0.04 | Income | −2673 *** | −3.16 | 1.45–1.50 | 143 | 162 |
| | 0.01 | Poverty | 0.03 ** | 2.14 | 1.15–1.20 | 143 | 162 |
| | 0.02 | Food security | −0.17 *** | −3.10 | 1.25–1.30 | 143 | 162 |

Note: NNM = nearest neighbor matching; MMM = mahalanobis metric matching; ATT stands for the average treatment effect for the treated. The results are significant at ***, **, * at the 1, 5, and 10% levels, respectively.

For wheat, the principal food crop, the impact of localized floods is negative and significant, signifying that farmers who experience localized floods have less wheat yield, in the range of 87.2–107.2 kg per acre. Such a substantial decrease in the yield of the principal food crop has grave implications for the food security of the farmers in Pakistan.

In this analysis, three different balancing tests are employed to check the matching quality: median absolute bias before and after matching, the value of R-square before and after matching, and the joint significance of covariates before and after matching and the result is presented in Table 7. The median absolute bias is quite high before matching and is in the range of 16.32–23.18. After matching, a considerable amount of bias has been reduced, and the bias is in the range of 4.3–6.2. Similarly, another test employed is the value of R-square before and after matching. The value is quite high before matching and is quite low after matching, indicating that after matching, both groups are almost similar. The joint significance of covariates indicates that there are systematic differences between both groups, but after matching both groups are very similar to each other. The indicators of covariates balancing are also presented in Figure 2. The results are inline with previous studies [46–49]. These studies also reported that after matching considerable amount of bias has been reduced and also the value of R-square is quite low after matching and also the joint significance of covariates should always been rejected after matching and should never be accepted before matching. This indicates that, after matching, both the categories of the farmers, i.e., affected through localized floods and not affected through localized floods, are very much similar to each other and there are no systematic differences between them.

**Table 7.** Indicators of covariate balancing (before and after matching).

| Matching Algorithm | Outcome | Median Absolute Bias before Matching | Median Absolute Bias after Matching | Percentage Bias Reduction | Value of R-Square before Matching | Value of R-Square after Matching | Joint Significance of Covariates before Matching | Joint Significance of Covariates after Matching |
|---|---|---|---|---|---|---|---|---|
| NNM | Wheat Yield | 20.15 | 5.23 | 74 | 0.234 | 0.003 | 0.002 | 0.278 |
| | Maize Yield | 21.30 | 6.20 | 71 | 0.337 | 0.004 | 0.003 | 0.283 |
| | Rice Yield | 20.04 | 4.73 | 76 | 0.264 | 0.003 | 0.002 | 0.291 |
| | Income | 23.18 | 5.28 | 77 | 0.421 | 0.001 | 0.001 | 0.265 |
| | Poverty | 18.54 | 4.61 | 75 | 0.390 | 0.004 | 0.002 | 0.248 |
| | Food Security | 16.32 | 5.29 | 68 | 0.324 | 0.003 | 0.003 | 0.234 |
| MMM | Wheat Yield | 22.59 | 4.30 | 81 | 0.428 | 0.002 | 0.002 | 0.178 |
| | Maize Yield | 21.43 | 5.26 | 75 | 0.365 | 0.003 | 0.003 | 0.172 |
| | Rice Yield | 20.45 | 4.79 | 77 | 0.379 | 0.004 | 0.002 | 0.153 |
| | Income | 19.34 | 5.12 | 74 | 0.410 | 0.003 | 0.003 | 0.162 |
| | Poverty | 18.34 | 5.13 | 72 | 0.234 | 0.002 | 0.002 | 0.153 |
| | Food Security | 20.53 | 5.12 | 75 | 0.419 | 0.004 | 0.003 | 0.148 |

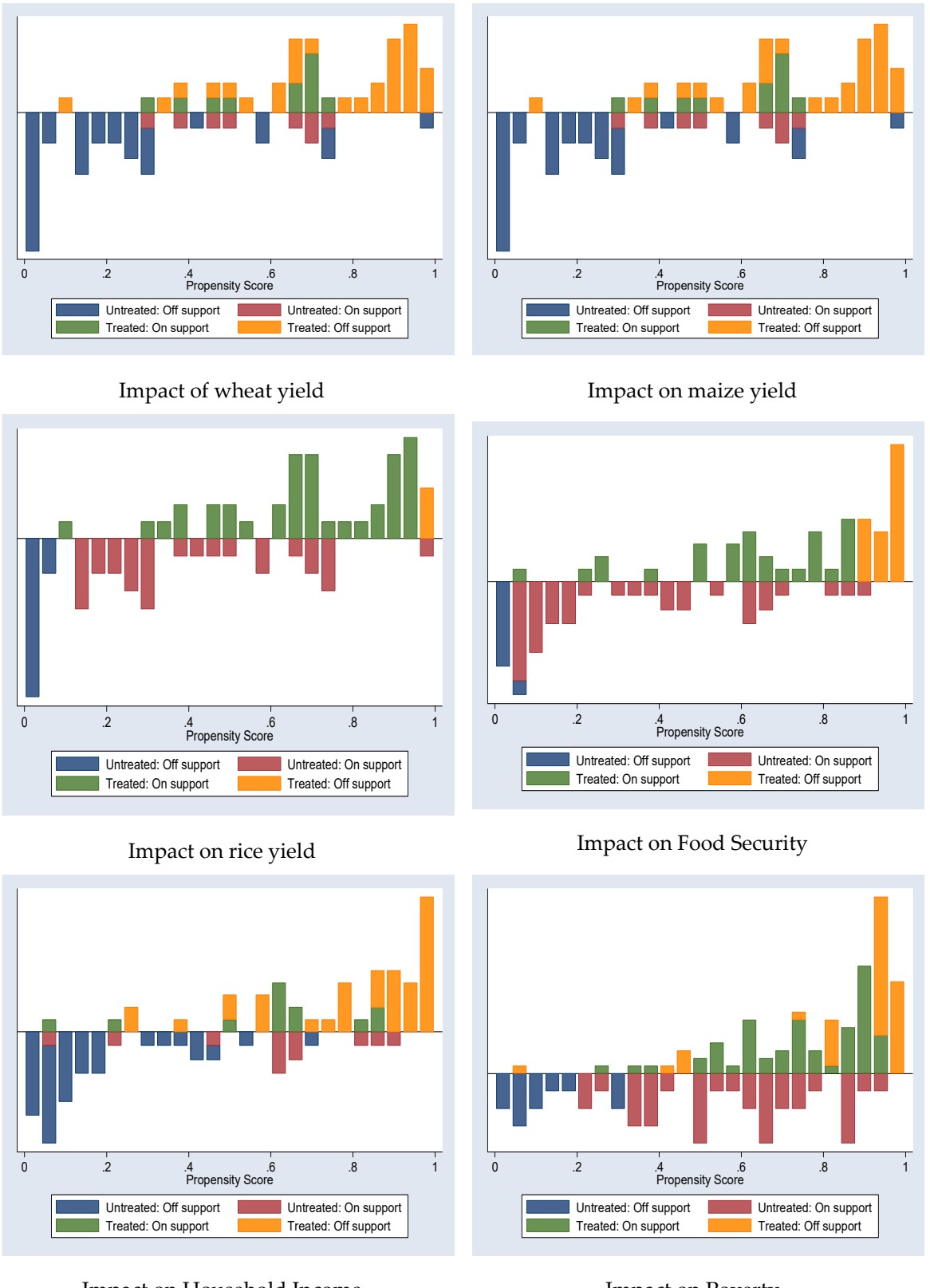

**Figure 2.** Indicators of covariate balancing before and after matching.

### 5. Conclusions and Policy Recommendation

Using primary data sets collected from all provinces of Pakistan, this paper analyzes the determinants of the adoption of strategies for coping with localized floods and the impact of localized floods on farmers' income, poverty, wheat, maize, and rice yield, and food security. We used a multivariate probit model to assess the determinants and PSM to investigate the impact of localized floods. Crop and livestock insurance, bund-making, land-leveling, and tree plantation were the primary strategies adopted by farm households to cope with localized floods. This study found that education and wealth were the most important drivers for the adoption of these strategies. More prosperous and educated households are more likely to adopt strategies to cope with localized floods. Localized floods have an adverse impact on the well-being and the livelihoods of rural farm households. The empirical results show that localized floods reduce the yield, food security, and income, and increase poverty levels. Hence policy should enhance the farm household capability to cope with localized floods. Among the different coping strategies, the trees plantation has been judged as the number one strategy, followed by crop and livestock insurance, land-leveling, and bund-making, respectively.

National-level floods attract the attention of various stakeholders, while localized floods are mostly unnoticed. Due to localized floods, maize yields are lowered, in the range of 2.53–3.11 maunds per acre. Rice yields are also decreased, in the range of 2.07–2.015 maunds per acre. Rice is an important food and cash crop, and this lowered yield has severe implications for rural households' livelihoods and food security levels. Although localized floods can have implications for all categories of farmers, this is especially the case for small-scale farmers as they are more vulnerable and have less extensive landholding and fewer assets.

From the empirical findings, it can be concluded that to minimize the impact of localized floods, the flood risk management practices need to be scaled out among the farming community through the agricultural extension department as well as other departments. Further research needs to explore the role of social safety nets, such as crop and livestock insurance schemes, in mitigating the risk of localized floods. Through the engagement of community-based organizations, participatory approaches need to be adopted to combat the risks of localized floods. As tree plantation has been regarded as the best mitigating strategy, campaigns need to be initiated to plant more trees to reduce the risk of localized floods.

**Author Contributions:** Conceptualization, A.A.and D.B.R.; methodology, A.A.and D.B.R..; software, Akhter Ali.; validation, A.A., and D.B.R..; formal analysis, A.A.; data curation, A.A.; writing—original draft preparation, A.A.and D.B.R..; writing—review and editing, A.A.and D.B.R. All authors have read and agreed to the published version of the manuscript.

**Funding:** This research was funded by CRP Maize-Agro Food System (CRP-Maize AFS) and Agricultural Innovation Program for Pakistan (AIP) Project funded by USAID.

**Conflicts of Interest:** The authors declare no conflict of interest.

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
