# Peer review of "Localized Floods, Poverty and Food Security: Empirical Evidence from Rural Pakistan"

_hydrology, doi:10.3390/hydrology7010002_

Round 1

Reviewer 1 Report

The current study provides a description of available flood mitigation techniques (crop and livestock insurance, bund-making, land-leveling and tree plantation) and describes the current situation of rural farm households in Pakistan. However, the study could do more to provide estimates about which mitigation technique is most beneficial for which kind of household, and which category of farm households could most benefit from mitigating flooding.

Comments:

1. Line 31: Be more specific: how much over how many years?

2. Lines 38 and 39: Be more specific (provide at least one estimate for an increase).

3. Line 47: Cars are not structures, you could say "structures and property".

4. Line 75 (Table 1): Provide a detailed source for the “Pakistan Weather Portal” reference (website, report, or personal communication).

5. Line 77: "2015" is listed as “2014” in the table.

6. Line 95: Explain acronyms like “KP”.

7. Line 105: Are villages that were damaged multiple times included only once or multiple times in the number 197,275? Now it sounds like they are included only once.

8. Line 212: Add description about the variables in eq. 1. Furthermore, eqs. 1 to 4 are misleading, because they assume that there is an iterative process whereby techniques such as bund-making are only adapted after crop and livestock insurance is in place. I’d recommend removing eqs. 1 to 4, because the paper doesn’t benefit from them and their meaning is explained multiple times in the manuscript.

9. Line 245 (Table 2): Add description of what the numbers in parentheses mean. Also, here you use (KPK), before it was (KP). Please be consistent.

10. Line 260ff: Use PSM instead of “propensity score matching” through out the text after you defined the acronym.

11. Line 274ff: Don’t mix “farmers”, “households”, and “frequency”. Keeping the same term for sections 2 and 3 and tables 2 and 3 makes it easier to understand the study. Repeat the number 812 in section 3, so the percentage is clearer. 

12. Line 322ff: Section 3.2.1 overlaps with sections 2.3 and 2.4. Make sure to not be repetitive. 

13. Line 381ff: Provide references for the matching algorithms NNM and MMM, and for ATT.

14. Lines 399 and 400: You can include the actual high and low values here again.

15. Line 434: Discuss similarities and differences between your findings and study [46].

Author Response

Reply to Reviewer 1

Dear Professor,

Greetings from CIMMYT

First of all we are grateful for all the valuable feedback, comments and suggestions from your side. We fully agreed with the comments and have tried to address them fully. The comments/suggestions have really helped us to improve the quality of our paper and we are grateful for that. The step wise reply to the comments is as follows;

Comments and Suggestions for Authors

The current study provides a description of available flood mitigation techniques (crop and livestock insurance, bund making, land leveling and tree plantation) and describes the current situation of rural farm households in Pakistan. However, the study could do more to provide estimates about which mitigation technique is most beneficial for which kind of household, and which category of farm household could most benefit from mitigating flooding.

Reply

Many thanks for this very important point, we have included details that which strategy can benefit which category of the farmers.

Comment 1

Line 31: Be more specific: how much over how many years?

Reply

Many thanks for this point, as suggested correction made.

Comment 2

Lines 38 and 39: Be more specific (provide at least one estimate for an increase).

Reply

As indicated, corrections made.

Comment 3

Line 47: Cars are not structures, you could say “structures and property”.

Reply

Many thanks for this point, correction made.

Comment 4

Line 75 (Table 1): Provide a detailed source for the “Pakistan Weather Portal” reference (website, report, or personal communication).

Reply

As suggested, “Pakistan Weather Portal” reference included.

Comment 5

Line 77: “2015” is listed as “2014” in the table

Reply

Many thanks, as suggested correction made.

Comment 6

Line 95: Explain acronyms like “KP”

Reply

We are grateful for suggesting this, the acronyms “KP” has been explained.

Comment 7

Line 105: Are villages that were damaged multiple times included only once or multiple times in the number 197,275? Now it sounds like they are included only once.

Reply

Many thanks for this point, yes they were included only once.

Comment 8

Line 212: Add description about the variables in Equation 1. Furthermore, equation 1 to 4 are misleading, because they assume that there is an iterative process whereby technique such as bund-making are only adapted after crop and livestock insurance is in place. I would recommend removing equations 1 to 4, because the paper doesn’t benefit from them and their meaning is explained multiple times in the manuscript.

Reply

Many thanks for this very important point. As suggested we have removed equation 1-4 and the variables in the equation has also been explained.

Comment 9

Line 245 (Table 2): Add description of what the numbers in parenthesis mean. Also, here you use (KPK), before it was (KP). Please be consistent.

Reply

We are grateful for this point. The details of the number in parenthesis are included in the footnote. For KP are corrections are made throughout the text.

Comment 10

Line 260 ff: Use PSM instead of “propensity score matching” through out the text after you defined the acronym

Reply

We are grateful for this point, as suggested we have used PSM instead of “Propensity score matching” throughout the text.

Comment 11

Line 274 ff: Don’t mic “farmers”, “households” and “frequency”. Keeping the same term for sections 2 and 3 and tables 2 and 3 makes it easier to understand the study. Repeat the numbers 812 in section 3, so that the percentage is clearer.

Reply

Many thanks for this very important suggestion. We have tried to be consistent and corrections have been made throughout the text.

Comment 12

Line 322 ff: Section 3.2.1 overlaps with section 2.3 and 2.4. Make sure to not be repetitive.

Reply

Many thanks for this comment, we have tried to correct the repetition.

Comment 13

Line 381ff: Provide references for the matching algorithms NNM and MMM and for ATT.

Reply

As suggested we have included references for the NNM, MMM and ATT.

Comment 14

Lines 399 and 400: You can include the actual high and low values here again.

Reply

Many thanks, as suggested we have included the actual high and low values

Comment 15

Line 434: Discuss similarities and differences between your findings and study (46)

Reply

Many thanks, as suggested we have discussed the similarities and differences between study and findings.

Once again we are grateful for all the very helpful comments on our paper. We have tried to incorporate the comments fully. The comments have really helped us to improve the quality of our paper and we are grateful for that.

Reviewer 2 Report

In the introduction, start the paragraphs too many times with the words Floods. Too repetitive!

The authors should emphasize the description of the study area, since they hardly mention the geographical and hydrological characteristics of Pakistan. It would be very interesting to add a map of altitudes from the Digital Terrain Model, in which the case study points and the hydrographic network of the country are shown, because with the map that the authors have made it is not clear.

In table 1: "Incidence of floods in Pakistan during the last decade", the authors can put in the last two decades, because the data of the reference are from the 2000s and 2010s.

In the discussion, the authors could be compare the results obtained with other works (reference is hardly made to other works!). Only in the last line and also just reference an article. “The PSM findings are inline with previous similar studies [46]”

The conclusion is very poor, and hardly collects information on the results, and does not raise proposals for improvement and future work proposals.

Author Response

Dear Professor,

Greetings from CIMMYT

First of all we are grateful for all the very helpful comments from your side. The comments have really helped us to improve the quality of our paper and we are grateful for that. We fully agreed with the comments and have tried to address them fully.

The step wise reply to the comments is as follows;

Comment 1

In the introduction, start the paragraphs too many times with the words Floods. Too repetitive

Reply

We are grateful for this point, as suggested we have made the corrections.

Comment 2

The authors should emphasize the description of the study area, since they hardly mention the geographical and hydrological characteristics of Pakistan. It would be very interesting to add a map of altitudes from the Digital Terrain Model, in which the case study points and the hydrographic network of the country are shown, because with the map that the authors have made it is not clear.

Reply

Many thanks for this point, as suggested we have included the revised map.

Comment 3

In the table 1: “Incidences of floods in Pakistan during the last decade”, the authors can put in the last two decades, because the data of the reference are from the 2000s and 2010s.

Reply

We are grateful for this suggestions, as suggested the correction has been made.

Comment 4

In the discussion, the authors could be compare the results obtained with other works (reference is hardly made to other works). Only in the last line and also just reference an article. “The PSM findings are inline with previous studies [46].

Reply

Many thanks for this very important point, as suggested we have compared the findings with previous work and more references has been included.

Comment 5

The conclusion is very poor, and hardly collects information on the results, and does not raise proposals for improvement and future work proposals.

Reply

As suggested we have tried to strengthen the conclusion section.

Once again we are grateful for all the very helpful comments and suggestion. The comments have really helped us to improve the quality of our paper and we are grateful for that.

Round 2

Reviewer 1 Report

Although the authors have addressed most of my previous concerns, my main concern remains that the study could do more to provide estimates about which mitigation technique is most beneficial for which kind of household, and which category of farm household could most benefit from mitigating flooding. Although the authors responded that this has been addressed in the revised manuscript, I could not find any additions towards suggesting for what region and type of farm it is most beneficial to adapt new measures, and which of the four (crop and livestock insurance, bund making, land leveling or tree plantation) seems to provide best protection, when compared with unprotected neighbors. This can be added especially in the abstract or conclusions.

Other remaining issues are my previous comments 4, 7, 13, and 15:

4. Provide a detailed source for the “Pakistan Weather Portal” reference (list in which website, report, or personal communication you found this information).

7. Can you add a clarification that villages that were damaged multiple times are included only once in the number 197,275? The current sentence structure made me ask if they are included only once.

13. Provide references for the matching algorithms NNM and MMM and for ATT and add them to the reference list, so the reader can understand how these methods work.

15. Discuss similarities and differences between your findings and study (46). It seems you just deleted the reference in the revised version.

Author Response

Reply to the Comments Dear Professor, Greetings First of all we are grateful to you and the anonymous reviewers for the very helpful comments on our paper. The comments have really helped us to improve the quality of our paper and we are grateful for that. We are also grateful for the second round of comments from one of the reviewers. We fully agreed with all the comments and have tried to address them fully. The stepwise reply to the comments is as follows; Overall comments Although the authors have addressed most of my previous concerns, my main concern remains that the study could do more to provide estimates about which mitigation technique is most beneficial for which kind of household, and which category of farm household could most benefit from mitigating flooding. Although the authors responded that this has been addressed in the revised manuscript, I could not find any additions towards suggesting for what region and type of farm it is most beneficial to adapt new measures, and which of the four (crop and livestock insurance, bund making, land leveling or tree plantation) seems to provide best protection, when compared with unprotected neighbors. This can be added especially in the abstract or conclusions. Reply First of all we are grateful for all the very helpful comments and suggestion. The comments have really helped us to improve the quality of our paper. Now we have included the details about various mitigating strategies i.e. which mitigating strategy is better as compared to other. The tree plantation has been ranked as the best mitigating strategy followed by crop and livestock insurance, land leveling and bund making respectively. The farming category wise analysis indicated that adoption of these flood mitigating measures s beneficial to all categories of the farmers especially to the small farmers as they are more vulnerable and have limited capacity to cope with these challenges. Comment Other remaining issues are my previous comments 4, 7, 13, and 15: 4. Provide a detailed source for the “Pakistan Weather Portal” reference (list in which website, report, or personal communication you found this information). Reply Many thanks for this very important point, as suggested we have included the website of “Pakistan Weather Portal.” Comment 7. Can you add a clarification that villages that were damaged multiple times are included only once in the number 197,275? The current sentence structure made me ask if they are included only once. Reply Yes, the villages were included only once and we have tried to rephrase the sentence to make it more clear. Comment 13. Provide references for the matching algorithms NNM and MMM and for ATT and add them to the reference list, so the reader can understand how these methods work. Reply As suggested we have included the references in the reference list for NNM, MMM and ATT and the previous studies having employed these matching algorithms has been cited. Comment 15. Discuss similarities and differences between your findings and study (46). It seems you just deleted the reference in the revised version. Reply Many thanks for highlighting this point. We have included more references and also have tried to discuss the similarities and differences between this study and the previous studies. Once again we are grateful for all the valuable comments and suggestions. The feedback has really helped us to improve the quality of our paper and we are grateful for that. Now hopefully the paper is in much improved form.
